# Enhancing Wheat Sprout Attributes Using "Intensification of Vaporization by Decompression to the Vacuum", an Innovative Drying–Texturizing Technology

Helga Francis [1], Espérance Debs [2], Richard G. Maroun [1] and Nicolas Louka [1,*]

1 Centre d'Analyses et de Recherche, UR Technologies et Valorisation Agroalimentaires Laboratoire d'Intensification des Procédés Industriels, Faculté des Sciences, Université Saint-Joseph, Beyrouth 1104 2020, Lebanon; helga.francis@usj.edu.lb (H.F.); richard.maroun@usj.edu.lb (R.G.M.)
2 Department of Biology, Faculty of Arts and Sciences, University of Balamand, Tripoli P.O. Box 100, Lebanon; esperance.debs@balamand.edu.lb
* Correspondence: nicolas.louka@usj.edu.lb; Tel.: +961-1421386

**Abstract:** Highly valued for their nutritional benefits, sprouts are characterized by high water content, which promotes microbial proliferation, potentially leading to toxicity and a reduced shelf life. To address this challenge, the present study explores the application of a novel drying–texturizing approach, named IVDV (Intensification of Vaporization by Decompression to the Vacuum), to sprouts. This technique would enable faster drying of the sprouts and better preservation of their nutritional content, compared to traditional hot-air drying. Using Response Surface Methodology, optimal IVDV parameters (saturated steam pressure P, processing time t, and water content W) for wheat sprouts were defined, with a focus on preserving vitamins, proteins, and lipids, and optimizing the expansion ratio. This optimization process identified optimal experimental conditions at 5.5 bars, a duration of 15 s, and 8.8% d.b. water content. Under these conditions, the use of IVDV endowed the expanded sprouts with a crunchier and more friable texture. In addition, it significantly improved the preservation of vitamins B6 and E by 412% and 42%, respectively, compared to traditional mild hot-air drying, without significantly affecting vitamin B2, proteins, and lipids. When combined with conventional hot-air drying, IVDV not only enhanced the preservation of the sprouts' nutritional content but also reduced drying time and energy consumption. This marks a significant advancement in sprouts preservation techniques, paving the way for novel potential applications.

**Keywords:** sprouts; wheat; hot-air drying; IVDV; vitamins; expansion

## 1. Introduction

Sprouts feature among today's food trends, with expectations for the market to continue growing in the coming years, driven by the increasing consumer demand for healthy and nutritious foods [1,2]. These greens are a valuable addition to any diet due to their high nutritional content and potential health benefits. In fact, sprouts are a good source of nutrients such as vitamins, minerals, fibers, and antioxidants, while being low in calories and fat [3–5]. They are also associated with numerous health benefits, such as improved digestion, immune function, and heart health [6,7].

While a variety of seeds, grains, and legumes, including radish, alfalfa, broccoli, mung beans, rice, and wheat, can be cultivated into sprouts, our study focused specifically on wheat. In fact, wheat is a fundamental food crop that enjoys global consumption and serves as a cornerstone ingredient in numerous dishes [8]. Specifically, wheat sprouts are renowned for their nutritional merits, which encompass a wide range of vitamins, phenolic compounds, free amino acids, dietary fibers, and minerals, thus presenting a beneficial and attractive addition to a balanced diet [3,9]. Notably, research on wheat sprouts has revealed higher levels of vitamins (B1, B2, B6, E, and β-carotene) compared

to ungerminated seeds [10–12] and increases both in total phenolic compounds and of the antioxidant capacity of wheat sprouts during germination [9,13].

To ensure the best quality and nutritional value, it is commonly advised that sprouts be consumed when they are in their fresh form. Nevertheless, fresh sprouts have been implicated in numerous instances of foodborne illnesses, with over 60 outbreaks reported from 1988 to 2020 [14]. The moist and warm conditions necessary for germination provide an optimal environment for bacterial proliferation. As a result, sprouts are prone to rapid spoilage, leading to a reduced average shelf life of approximately 3 to 5 days.

In order to address this issue arising from the presence of free water within the product, two primary preservation methods are employed: freezing, which seeks to immobilize the water, rendering it inaccessible to bacteria, and drying, aimed at removing the majority of the water content. However, conventional freezing can modify the product's structure, thereby diminishing its market value, with the exception of rapid freezing, which incurs prohibitive costs [15–18]. Alternatively, drying is frequently performed using various techniques, notably lyophilization and hot-air methods. Lyophilization is recognized as the superior method for preservation, though its applications are typically limited to products of high sensitivity and substantial value, which can be attributed to the considerable expenses involved [19]. Conversely, while air drying, the oldest and most widespread preservation method, effectively extends the shelf life of products, it often results in alterations to their physical form and a diminution of their nutritional content [20].

Overall, research efforts on sprout drying techniques are scarce and incomplete. Certain studies have drawn comparisons between solar and hot-air drying, focusing on nutritional parameters such as folate content or antioxidant activity [21,22]. Several studies have investigated various temperatures in hot-air drying processes [23], while only a few have compared these to freeze drying [24–26].

The prevailing consensus emerging from these research findings suggests that mild hot-air drying, conducted at temperatures ranging from 40 to 60 °C, is the recommended choice for preserving sprouts [24,27]. Nonetheless, a low drying temperature results in a prolonged air-drying duration. The free water remaining within the product following the first stage of drying is often trapped in its compacted dense structure, requiring increased energy and heat for its elimination. Obviously, this extended process contributes to greater thermal degradation of the sprouts.

To tackle this issue, researchers have investigated the impact of grinding on the drying kinetics of sprouts, demonstrating that this pretreatment method effectively shortens the drying duration [24]. Yet this destructive technique is solely feasible when the desired end product is in the form of powder or small flakes. Applying "Intensification of Vaporization by Decompression to the Vacuum" (IVDV), a texturizing technique designed and patented by our team (Patent reference: 2019/05-11666L), promises a compelling approach for the preservation of wheat sprouts. This technology consists of a thermo-mechanical treatment using saturated steam pressure. It provides a fast pressurization, reaching 12 bars in less than one second, by means of an ultra-rapid steam generation system. This is followed by a sudden drop to the vacuum after a stay of a few seconds at this high pressure, resulting in the product's expansion [28–30]. IVDV proved to be efficient in several processes, such as defatting of peanuts [28], expansion of maize [30], and extraction of biomolecules from olive leaves [29]. Similar to its precursor process, the Instantaneous Controlled Pressure Drop (DIC) technology [31], which has been validated for its efficacy in decontaminating various food products (including baby food and powders) [32], the IVDV technique also exhibits a decontamination effect. It also enhances the preservation of heat-sensitive products and active compounds, making it ideal for sprouts. In fact, combining this texturizing technique with a mild hot-air drying would expand the sprouts and facilitate the residual water extraction, thus accelerating the drying kinetics and alleviating thermal degradation.

The primary aim of this study is to evaluate the effects of integrating an IVDV treatment within a conventional mild hot-air drying cycle applied to wheat sprouts. Further-

more, operational parameters influencing the nutritional and textural properties of the sprouts will be optimized in order to improve their overall quality.

## 2. Materials and Methods

### 2.1. Chemicals

Diethyl ether used for the crude fat assay was purchased from Scharlau (Barcelona, Spain), while all other chemicals, solvents and HPLC standards were provided by Sigma-Aldrich Chemical Co. (St. Louis, MO, USA). All reagents used were of analytical or HPLC grade when needed.

### 2.2. Germination Process

Seeds of organic whole soft winter wheat (*Triticum aestivum* L.) were acquired from Naturalia s.a.l. (Baabda, Lebanon).

The germination process for these seeds was established in our previous work [33]. Briefly, wheat seeds were soaked for 3 h in mineral water with a mass-to-volume ratio of 1:5 (g/mL) and a soaking-water renewal every hour. Afterwards, drained seeds were left to germinate at 22 °C ± 1 °C for 7 days in a sterile plastic box whose bottom was covered with humidified Whatman No. 1 filter papers.

These fresh sprouts, comprising the radicle and the initial shoot emerging from the seed, are then evenly spread over rectangular metallic mesh trays, with each tray containing approximately 200 g. The initial water content of these sprouts averaged around 350% d.b.

### 2.3. Hot-Air Drying and IVDV Treatment

The process that we are considering for the drying of sprouts is a mild hot-air drying process coupled with a treatment by IVDV. It consists of an initial partial hot-air drying in order to bring the product to the pre-IVDV water content required for processing, followed by the IVDV treatment itself and then the final drying, bringing the water content of the product to around 5%, which is ideal for long preservation [23].

Partial and final hot-air drying treatments for the sprouts were performed at 50 °C in an UFE 700 ventilated oven (Memmert GmbH, Schwabach, Germany) with air circulation of 1 m/s. The complete experimental protocol is outlined in Figure 1.

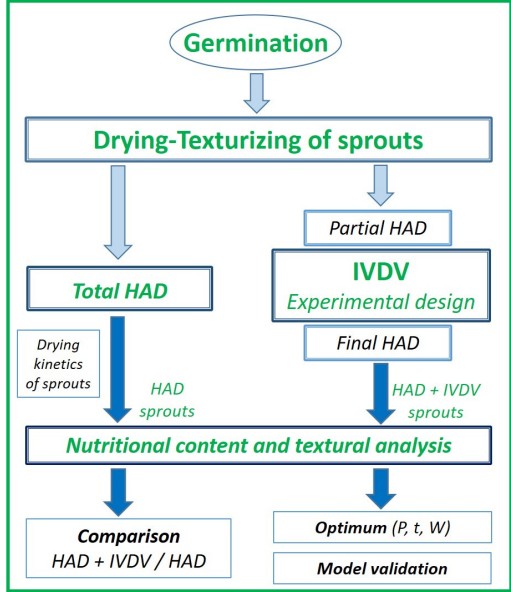

**Figure 1.** Work stage chart: HAD (Hot-Air Drying); IVDV (Intensification of Vaporization by Decompression to the Vacuum); P (saturated steam pressure), t (processing time), and W (water content before IVDV).

The IVDV processor features, as illustrated in Figure 2, allow for a thermo-mechanical treatment. The partially dehydrated product is placed in the treatment chamber (Figure 2(1)) and subjected to a saturated steam pressure of up to 12 bars. The reactor also includes a vacuum tank, a vacuum pump, a steam generator and an ultra-speed pressure-increase system (EPUR), as described in Figure 2.

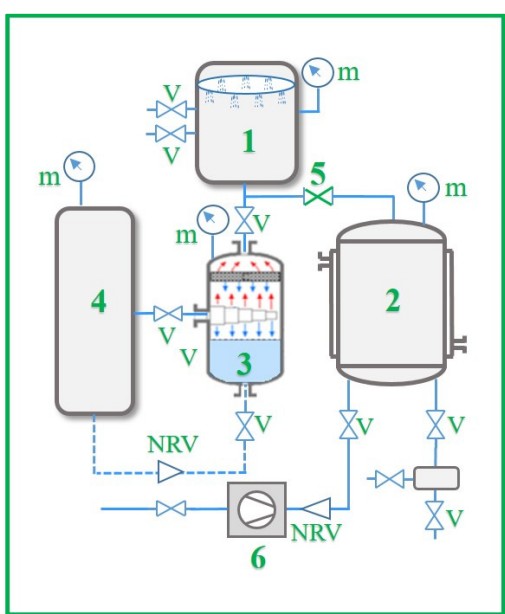

**Figure 2.** Schematic diagram of the IVDV apparatus: 1 (treatment chamber); 2 (vacuum tank); 3 (rapid steam generator); 4 (boiler); 5 (decompression valve); 6 (vacuum pump); m (manometer); NRV (non-return valve); and V (valve).

The treatment takes place in 5 stages:

1. Once the partially dried sprouts are introduced into the treatment chamber, a preliminary vacuum is established, allowing a direct contact between the surface of the product and the water vapor.
2. An ultra-rapid steam generator creates a rapid increase in saturated steam pressure (in less than a second).
3. The samples are subjected to high pressure P of saturated steam (high temperature) for a short time t (few seconds), to the point at which the product reaches the thermal and rheological stages required for its expansion.
4. During this phase, a sudden drop in pressure towards vacuum induces a fast evaporation of a part of the residual water inside the product. Within the product, this induces mechanical stresses of viscoelastic behavior, leading to its expansion and giving it a porous structure. It also leads to a very rapid cooling of the product, which is accompanied by a simultaneous injection of atmospheric (ambient) air, while maintaining the vacuum. This considerably limits the thermal degradation of the product and makes it possible to avoid shrinkage by freezing it in its expanded state.
5. A return to atmospheric pressure is ensured before opening the treatment chamber and withdrawing the expanded product.

### 2.4. RSM Experimental Design

In order to optimize the IVDV treatment, Response Surface Methodology (RSM) was applied, using a rotatable central composite design with 20 runs. This design integrates 8 factorial design points (coded as −1 for low and +1 for high levels), 6 star points (coded as −α and +α), and 6 repetitions at the central level (coded as 0 for error estimation). This statistical and mathematical modeling tool allows the identification of the relationships between the input variables and the output response variables of the process. In our case,

the three independent variables affecting the treatment are the saturated steam pressure (P), the processing time (t), and the initial water content (W). Table 1, below, displays the levels of the variables, which were determined following preliminary analysis.

**Table 1.** Variables levels used for the experimental design for wheat sprouts.

| Variable | Low (−1) | Medium (0) | High (+1) |
|---|---|---|---|
| Pressure (bars) | 3 | 4.5 | 6 |
| Processing time (s) | 7 | 12 | 17 |
| Water content (% d.b.) | 15 | 25 | 35 |

The responses monitored were the expansion ratio (ER), the lipids content (Lip.), the proteins content (Prot.), and the vitamin B2 (Vit.B2), B6 (Vit.B6), and E (Vit.E) contents. The design of experiments and the statistical analysis of the results were conducted utilizing STATGRAPHICS Centurion XVI.I.

*2.5. Nutritional and Textural Analysis*

The **water content** was gravimetrically determined according to the AOAC method. Samples in triplicate were dried to a constant weight in an oven at 105 °C for 24 h, and their weight loss monitored [33].

**Crude fat and proteins** were analyzed following the corresponding AOAC methods, with minor modifications, as described in our previous work [33]. Vitamins B2, B6, and E were monitored by HPLC [33] using a liquid chromatography (Waters Alliance, Milford, MA, USA) system coupled to a control system and data collection software, namely, Empower 3.

**Expansion ratio** (ER) analysis is used to evaluate the textural quality of the sprouts after the drying–IVDV treatment. In our study, ER reflects the increase in volume of the IVDV-treated sprouts, compared to the HAD-dried ones of an equivalent weight. It was assayed following the Louka and Allaf [34] method by calculating the ratio of the specific density of the HAD + IVDV samples to that of the HAD sprouts.

**Drying kinetics** of sprouts were monitored by placing samples of sprouts in rectangular metallic mesh trays at 50 °C and weighing them at specific time intervals until constant weight ($\Delta m \leq 0.1$ mg). These intervals were: every 5 min for the first 20 min, then every 10 min from 20 to 60 min, every 20 min from 60 to 120 min, every 30 min from 120 to 240 min, and finally hourly up to 1440 min.

Their water content (W) was then calculated as follows:

$$W \ (\% \ \text{d.b.}) = m_W / m_{MS}$$

where

- $m_{MS}$, the mass of dry matter, is obtained by the determination of the initial water content after incubating the samples in triplicate in a ventilated oven at 105 °C for 24 h.
- $m_W$, the mass of water, is equal to the mass of the product at the time "t" minus the mass of dry matter ($m_{MS}$).

**3. Results and Discussion**

The experimental matrix with the actual levels of the processing parameters studied (P, t, and W) are presented in Table 2, along with the values of the different responses measured (Prot., Lip., Vit.B2, Vit.B6, Vit.E, and ER). Every assay was performed in triplicate to ensure the repeatability of the results, and the mean values are reported in Table 2.

**Table 2.** Experimental matrix for wheat sprout IVDV treatment: P (pressure); t (processing time); W (water content before IVDV treatment); Lip.: crude fat; Prot.: crude proteins; ER: expansion ratio; Vit.: vitamin; HAD: untreated hot-air dried samples.

| Run | Process Parameters | | | Response Parameters | | | | | |
|---|---|---|---|---|---|---|---|---|---|
| | P (bars) | t (s) | W (% d.b.) | Lip. (% d.b.) | Prot. (% d.b.) | Vit.B2 (mg/kg) | Vit.B6 (mg/kg) | Vit.E (mg/kg) | ER (%) |
| 1 | 3[−1] | 7[−1] | 15[−1] | 3.29 | 12.69 | 1.79 | 2.55 | 11.84 | 1.42 |
| 2 | 6[+1] | 7[−1] | 15[−1] | 3.50 | 11.98 | 2.13 | 3.89 | 13.80 | 1.35 |
| 3 | 3[−1] | 17[+1] | 15[−1] | 3.90 | 12.45 | 1.85 | 3.16 | 15.18 | 1.44 |
| 4 | 6[+1] | 17[+1] | 15[−1] | 3.33 | 11.74 | 2.09 | 3.79 | 13.86 | 1.58 |
| 5 | 3[−1] | 7[−1] | 35[+1] | 3.04 | 12.59 | 1.87 | 1.82 | 10.28 | 1.29 |
| 6 | 6[+1] | 7[−1] | 35[+1] | 3.54 | 12.51 | 2.05 | 3.40 | 14.63 | 1.47 |
| 7 | 3[−1] | 17[+1] | 35[+1] | 3.53 | 12.23 | 2.11 | 1.48 | 12.49 | 1.29 |
| 8 | 6[+1] | 17[+1] | 35[+1] | 3.20 | 11.19 | 1.95 | 2.71 | 11.68 | 1.58 |
| 9 | 1.98[−α] | 12[0] | 25[0] | 3.87 | 12.44 | 1.76 | 1.68 | 9.12 | 1.26 |
| 10 | 7[+α] | 12[0] | 25[0] | 3.42 | 11.45 | 2.18 | 3.73 | 12.75 | 1.48 |
| 11 | 4.5[0] | 3.6[−α] | 25[0] | 3.15 | 11.85 | 1.94 | 2.40 | 12.21 | 1.31 |
| 12 | 4.5[0] | 20.4[+α] | 25[0] | 3.61 | 11.50 | 1.99 | 2.40 | 15.54 | 1.49 |
| 13 | 4.5[0] | 12[0] | 8.2[−α] | 3.28 | 11.40 | 1.97 | 5.02 | 13.03 | 1.60 |
| 14 | 4.5[0] | 12[0] | 41.8[+α] | 3.19 | 12.29 | 2.02 | 2.68 | 12.97 | 1.41 |
| 15 | 4.5[0] | 12[0] | 25[0] | 3.53 | 10.09 | 2.04 | 2.46 | 14.42 | 1.47 |
| 16 | 4.5[0] | 12[0] | 25[0] | 3.33 | 10.36 | 2.02 | 2.24 | 14.79 | 1.44 |
| 17 | 4.5[0] | 12[0] | 25[0] | 3.49 | 10.65 | 2.03 | 2.02 | 14.62 | 1.41 |
| 18 | 4.5[0] | 12[0] | 25[0] | 3.69 | 10.16 | 2.07 | 2.93 | 14.86 | 1.45 |
| 19 | 4.5[0] | 12[0] | 25[0] | 3.58 | 10.31 | 2.15 | 2.09 | 14.71 | 1.46 |
| 20 | 4.5[0] | 12[0] | 25[0] | 3.61 | 10.55 | 2.05 | 2.69 | 14.68 | 1.47 |
| **HAD** | | | | 2.77 | 11.90 | 2.13 | 0.98 | 9.91 | 1.00 |

*3.1. Response Parameters*

3.1.1. Expansion Ratio

The IVDV treatment entails a rapid vacuum-induced expansion, generating mechanical constraints responsible for the alveolation and expansion of the treated product. Notably, across 20 runs, Expansion Ratio (ER) varied from 1.26% [run 9 (P = 1.98 bars; t = 12 s; W = 25%)] to 1.60% [run 13 (P = 4.5 bars; t = 12 s; W = 8.2%)], demonstrating consistent expansion.

Optical microscopy of wheat grains validated these findings, with Figure 3 illustrating that seeds treated with IVDV (Figure 3B) exhibited a greater volume compared to untreated seeds (Figure 3A). Moreover, observation of the cutouts indicated differential expansion and alveolation levels correlated with the treatment intensity, as shown in Figure 3D–G. Suboptimal treatment conditions led to partial gelatinization, characterized by the emergence of white and translucent areas, together with minimal alveolation (Figure 3E,G). The initially opaque product became more translucent with a partial fusion of the crystallites. Conversely, under treatment conditions approaching the optimum, consistent alveolation along with gelatinization is observed. Here, the initially crystalline product became amorphous and translucent (Figure 3D,F). These observations were corroborated by Maache-Rezzoug et al. [35], who showed a reduction in the crystallinity of wheat grain starch following the application of DIC, a process similar to the IVDV.

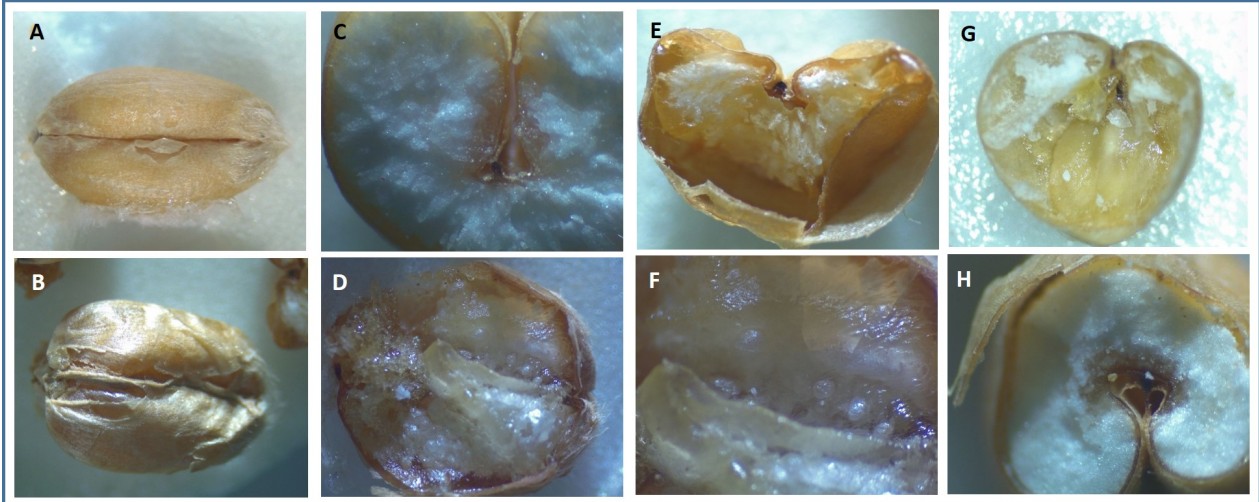

**Figure 3.** Wheat grains and sections viewed using a stereomicroscope (×35): (**A**,**C**) ungerminated seed; (**B**,**D**–**G**) sprouted seed treated by IVDV; and (**H**) sprouted seed dried by HAD.

In summary, these observations confirmed the expansion of the grains, reflecting the acquisition of a less compact product with a lighter and crunchier texture, compared to the initial grains (Figure 3A,C), or even the germinated grains that were totally dried by conventional hot-air drying (Figure 3H).

Based on the Pareto chart (Figure 4a), it can be noted that, with the exception of the interaction between processing time (t) and initial water content (W), all factors had a significant effect at a 95% confidence interval. Specifically, pressure and processing time exerted positive effects, as did the interactions of pressure with time (Pt) and water content (PW).

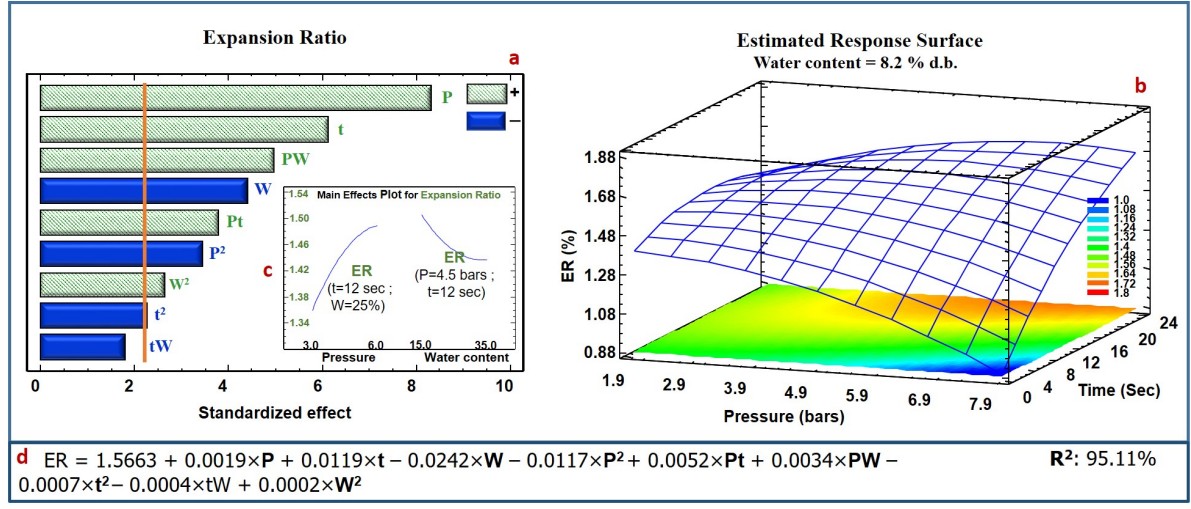

**Figure 4.** Experimental design results for Expansion Ratio in wheat sprouts. (**a**) Pareto charts, (**b**) Response surfaces, (**c**) Main effects plot, and (**d**) Regression model equations and determination factors.

### 3.1.2. Vitamins

It is essential to note that vitamin levels were measured in the final product (after treatment by IVDV and complete drying).

The vitamin levels in sprouts treated with IVDV ranged from 1.48 mg/kg (run 7) to 5.02 mg/kg (run 13) for vitamin B6, and from 9.12 mg/kg (run 9) to 15.54 mg/kg (run 12) for vitamin E. These concentrations remained higher than those in hot-air dried sprouts (0.98 mg/kg and 9.91 mg/kg for vitamins B6 and E, respectively), thus highlighting the

benefit of treatment by IVDV in preserving vitamins B6 and E. However, this effect was not observed for vitamin B2, with the maximum value obtained after treatment being similar to that of conventionally hot-air dried sprouts (2.18 mg/kg (run 10) and 2.13 mg/kg for complete drying).

### 3.1.3. Crude Proteins and Lipids

The protein content in treated wheat sprouts varied between 10.09% and 12.69%, while their lipid levels remained around $3.5 \pm 0.5\%$. Although there was a slight improvement in extractability under specific conditions (run 1), the IVDV treatment effectively preserved the levels of proteins and lipids, with no significant alterations found. Hence, the examination of these two responses will not be continued in later discussions.

### 3.2. Effects of Factors on Different Responses

In order to establish a model depicting the relationships between the operational process parameters and each of the responses, experimental values were analyzed using multiple regression analysis to fit a second-order polynomial equation. These equations are illustrated in Figures 4 and 5, where operational parameters with significant effects (linear and quadratic for each factor and the interaction between them) on each response parameter at a 95% confidence level ($p$-value < 0.05) are highlighted in bold. Table 3 compiles the $R^2$ correlation coefficients, adjusted $R^2$ values, and lack-of-fit data, denoting the model equation's significance and the degree of fit.

**Table 3.** Model fit indicators; $R^2$ correlation coefficients, adjusted $R^2$ values, and lack-of-fit data.

| Response Parameter | $R^2$ (%) | $R^2$ Adjusted (%) | Lack of Fit |
|:---:|:---:|:---:|:---:|
| ER | 95.1 | 90.7 | 0.15 |
| Vit.B2 | 88.9 | 80.0 | 0.33 |
| Vit.B6 | 93.8 | 88.3 | 0.83 |
| Vit.E | 90.5 | 82.0 | <0.05 |

In the evaluation of the model's fitness for vitamins B2, B6, and E, as well as the expansion ratio, a strong alignment with empirical data was observed, as evidenced by high $R^2$ and adjusted $R^2$ values across all parameters, indicating robust model performance. There was adequate fit for vitamins B2 and B6, and the expansion ratio was supported by lack-of-fit $p$-values greater than 0.05. However, vitamin E presents a contrast, with a lack-of-fit $p$-value below 0.05, suggesting an initial poor fitness. This perceived inadequacy has nuances to it. It is influenced by the method of assessment used by the software (Statgraphics) (https://www.statgraphics.com/), which compares the model's error to the experimental error. The notably low experimental error for vitamin E (0.15) relative to the model error (approximately 0.6) significantly impacts this calculation, though it may not indicate a genuine limitation in the model's predictive ability. Thus, the multiple regression model can be used to predict any combination of variables and explain the relationship between the process and the response within the specified measurement range.

The analysis of the outcomes from the 20 conducted runs allowed us to draw several conclusions regarding the impacts of the different factors in the IVDV treatment. However, it is noteworthy that a correlation has been identified between the effect of a parameter on the expansion ratio and its influence on the other responses. Consequently, in the following, we will primarily focus on interpreting the effect of each factor on the expansion.

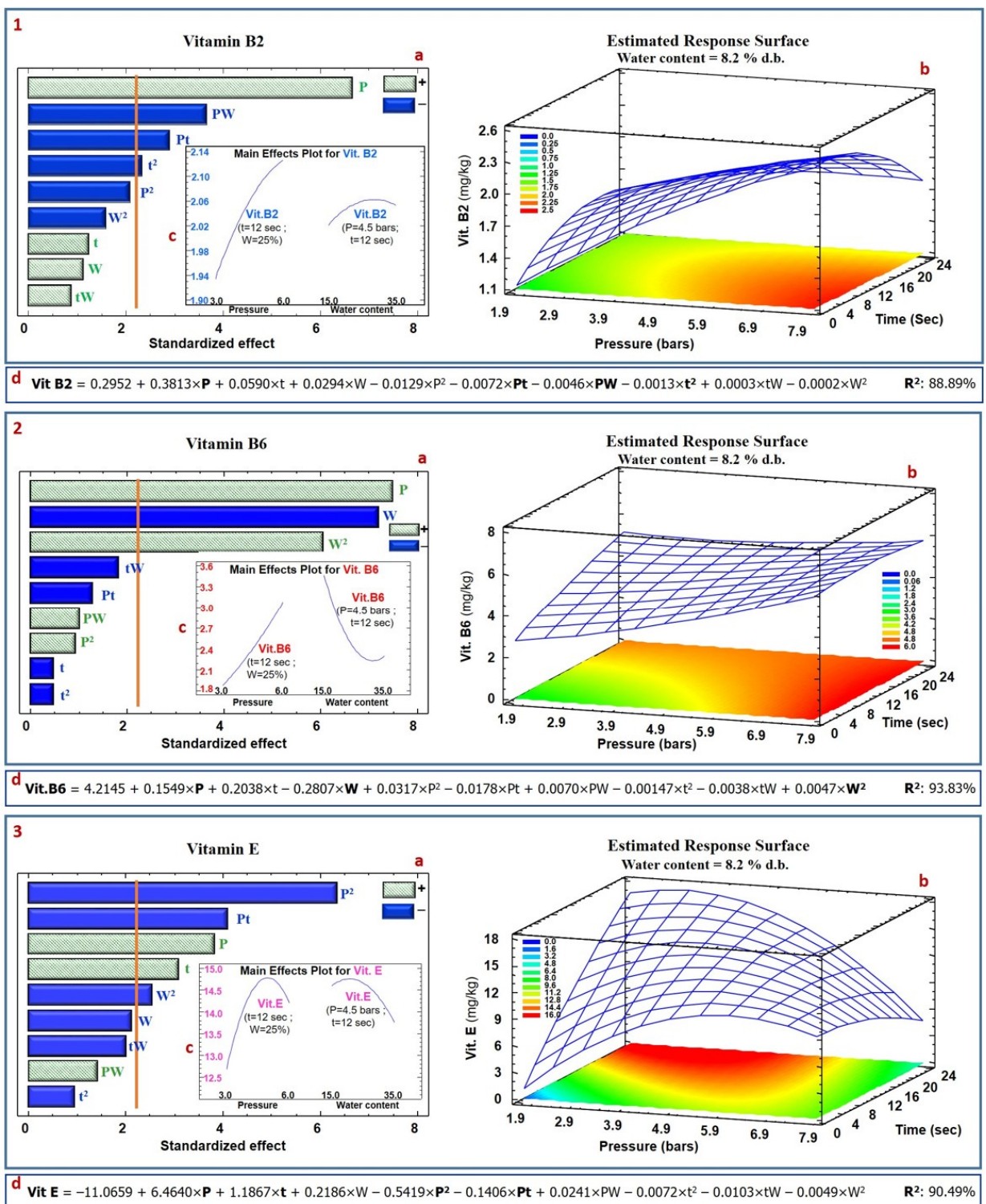

**Figure 5.** Experimental design results for vitamins B2 (**1**), B6 (**2**), and E (**3**) in wheat sprouts. (**a**) Pareto charts, (**b**) Response surfaces, (**c**) Main effects plot, and (**d**) Regression model equations and determination factors.

### 3.2.1. Effect of Saturated Steam Pressure (P)

Pareto diagram analyses (Figures 4a and 5(1a,2a,3a)) reveal that saturated vapor pressure (P) significantly enhanced the preservation of nutritional and textural qualities in dried sprouts, acting as the dominant factor for both expansion and vitamins B2 and B6

content. It is also a crucial variable in IVDV since it sets the treatment temperature and intensity [28,29], with increased pressure raising temperature and accelerating vitamin degradation reactions according to Arrhenius's law. However, the short IVDV treatment duration (rise in pressure in less than one second and average processing time of twelve seconds) prevented extensive vitamin loss. Furthermore, the IVDV treatment expanded the product, creating a porous structure (Figure 3) which facilitates faster water removal, thereby shortening the typically slow second phase of drying, and enhancing vitamin preservation. This approach effectively balanced the onset of degradation with accelerated drying, ensuring better preservation of the product's nutritional value.

### 3.2.2. Effect of Initial Water Content (W)

Our findings indicated that water content (W) had a significant negative effect on expansion ratio and the levels of vitamins B6 and E, with the positive effect for vitamin B2 being insignificant.

Research highlights that lower water content prior to IVDV enhanced product expansion more effectively than higher water content [28,34]. In fact, W had a dual effect: higher water content (at a given pressure and temperature) increased steam generation upon the release of pressure, resulting in the significant mechanical stresses that drive product expansion. In contrast, higher water content also enhanced product elasticity, leading to greater, but less stable, expansion; the product expanded significantly but retracted after treatment, reverting to its original form. This explained the observed negative impact of W on expansion shown in Figure 4. Furthermore, the plot of Expansion Ratio (ER) versus W (Figure 4c) at P = 4.5 bars and t = 12 s demonstrated a nearly linear relationship between these parameters, with stabilized ER for water contents beyond 30%.

Similarly, pre-IVDV water content generally had a negative effect on vitamin levels. The compacted state of the product post-expansion slows water removal and drying, increasing thermal degradation during the second drying phase, particularly for vitamin B6. Main effect plots revealed the following:

- For vitamin B2 (Figure 5(1)), W had no effect, with levels ranging only between 2.02 and 2.05 mg/kg.
- For vitamin B6, levels decreased from 3.46 to 2.30 mg/kg as water content increased from 15% to 25% (Figure 5(2)), a trend similar to that of the expansion. Afterwards, the levels of vitamin B6 stabilized, proving the positive quadratic effect of W.
- For vitamin E (Figure 5(3)), levels were initially stable, at around 14.6 mg/kg (for W between 15 and 28%), but decreased to 13.7 mg/kg as W reached 35%, correlating with a decrease in expansion. This reflected the negative quadratic effect of W.

Generally, vitamin levels followed similar trends to expansion, except for vitamin B2, which showed better resistance at the beginning of the treatment.

### 3.2.3. Effect of Processing Time (t)

Analysis of Pareto diagrams (Figures 4a and 5(1a,2a,3a)) of the various responses highlights that processing time predominantly exerted significant positive effects on product expansion and vitamin E content. Its negative impacts, such as on vitamin B6 levels, were non-significant. Treatment duration, in conjunction with temperature, critically determines treatment intensity, playing a pivotal role in product expansion. This was supported by findings indicating that a certain period was essential for initially "hard" products to become pliable enough for expansion. During this heating (and moistening) phase, the product rapidly achieved thermal equilibrium, attaining a viscoelastic behavior and the rheological state necessary for expansion [31]. This preparatory stage allowed the product to be inflated and subsequently maintain its expansion, or alternatively, to retract if elasticity was preserved.

Conclusively, our findings suggest that both product expansion and vitamin content were primarily influenced by the treatment pressure and duration. Among the factors evaluated, pressure emerged as the most influential, with responses being maximized under

high-pressure conditions. Treatment duration impacted outcomes in two ways: it either enhanced expansion, facilitating material transfer and product drying—thus better preserving vitamins—or led to vitamin destruction if the treatment was excessively prolonged.

### 3.2.4. Effects of Significant Interactions

- Effect of Pressure and Time Interaction (Pt)

The interaction between pressure and treatment duration (Figure 6a,c,d,f) collectively exhibited a positive effect on the expansion ratio. Short-duration treatments showed slight improvements, while longer durations led to more substantial expansions. However, this interaction negatively impacted vitamin B2 and E levels. This could be attributed to material expulsion during violent steam explosion and/or the degradation of vitamins by a relatively extended treatment under high temperature.

- Effect of Pressure and Water Content Interaction (PW)

The interaction between pressure and water content significantly enhanced the expansion ratio of the product, as indicated by the Pareto analysis (Figure 4) and demonstrated in the Expansion Ratio (ER) trends, with pressure variations for water contents of 15% and 35% (Figure 6b,e). ER exhibited negligible variation when the treatment was conducted at pressures of 3 or 6 bars. At a water content of 15%, an unvarying amount of water is evaporated irrespective of the applied pressure, leading to the expansion of the product with low elasticity, and it then retains its expanded state. At a water content of 35%, the volume of water evaporating is greater, resulting in a more pronounced initial expansion. The elasticity of the product being enhanced, this expansion is followed by a contraction, with the ER decreasing to 1.3% at a pressure of 3 bars and to 1.5% at a pressure of 6 bars.

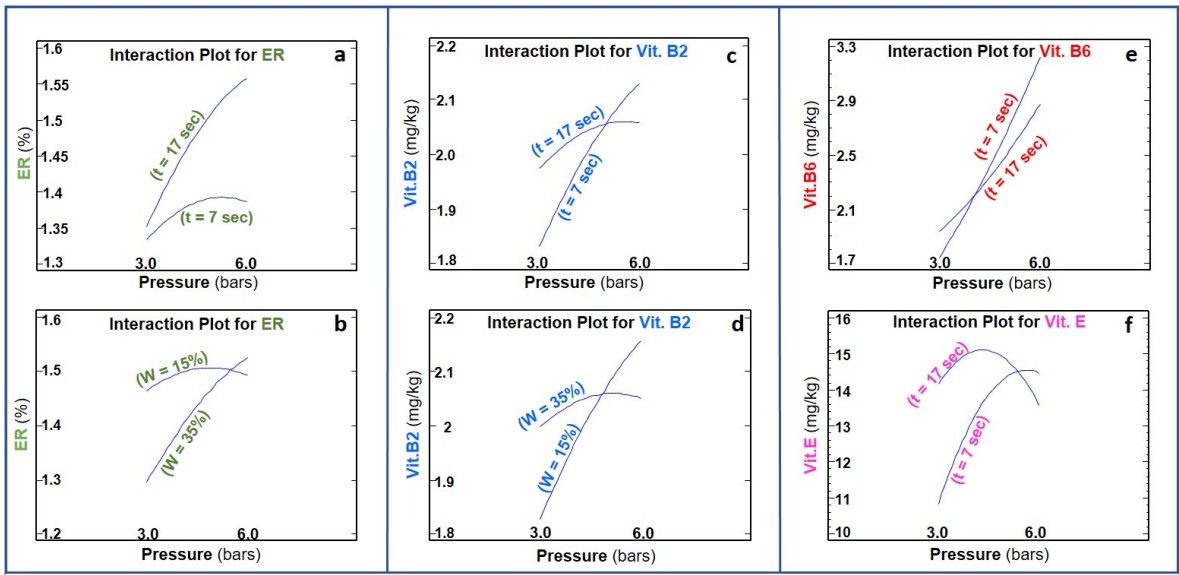

**Figure 6.** Significant interaction (Pt and PW) effects plots for Expansion Ratio (**a**,**b**); vitamin B2 (**c**,**d**); vitamin B6 (**e**); and vitamin E (**f**).

### 3.3. Optimization and Model Validation

- **Optimization:** The examination of results allowed us to define various processing conditions tailored to the particular response or responses to be enhanced. In single-response optimization, the objective was to focus on one particular response, enabling the selection of the best operational conditions for each response variable. Table 4, provided below, summarizes the optimal responses recommended by the Statgraphics processing software for each factor, along with the values of the related parameters.

**Table 4.** Optimal values of responses and corresponding parameters resulting from single and multiple optimization.

| | Response | Optimal Value | IVDV Conditions | | |
|---|---|---|---|---|---|
| | | | P (bars) | t (sec) | W (% d.b.) |
| Single optimization | ER (%) | 1.67 | 5.8 | 20.4 | 8.18 |
| | Vit.B2 (mg/kg) | 2.32 | 7.02 | 4.23 | 8.18 |
| | Vit.B6 (mg/kg) | 5.59 | 7.02 | 9.01 | 8.18 |
| | Vit.E (mg/kg) | 16.5 | 3.52 | 20.4 | 9.44 |
| Multiple optimization | ER (%) | 1.60 | | | |
| | Vit.B2 (mg/kg) | 2.05 | 5.48 | 15.4 | 8.8 |
| | Vit.B6 (mg/kg) | 5.02 | | | |
| | Vit.E (mg/kg) | 14.08 | | | |

Since we were seeking to maximize both the nutritional content and the expansion of the sprouts, we therefore opted for a total multiple optimization, taking into account the four response parameters of interest shown in Table 4. Subsequently, these conditions will be considered during the model validation.

Figure 7 shows the contour plots of each parameter at a set water content of 8.2% d.b. The colored regions correspond to the optimal areas for each of the factors alone. The optimal point, as defined by the software for multiple optimization, is represented by an asterisk, while the optimal region is outlined in yellow. As for the purple spot, it corresponds to a more restricted optimal area which could yield slightly more interesting response values, with ER = 1.64%, Vit.B2 = 2 mg/kg, Vit.B6 = 5.25 mg/kg, and Vit.E = 14 mg/kg.

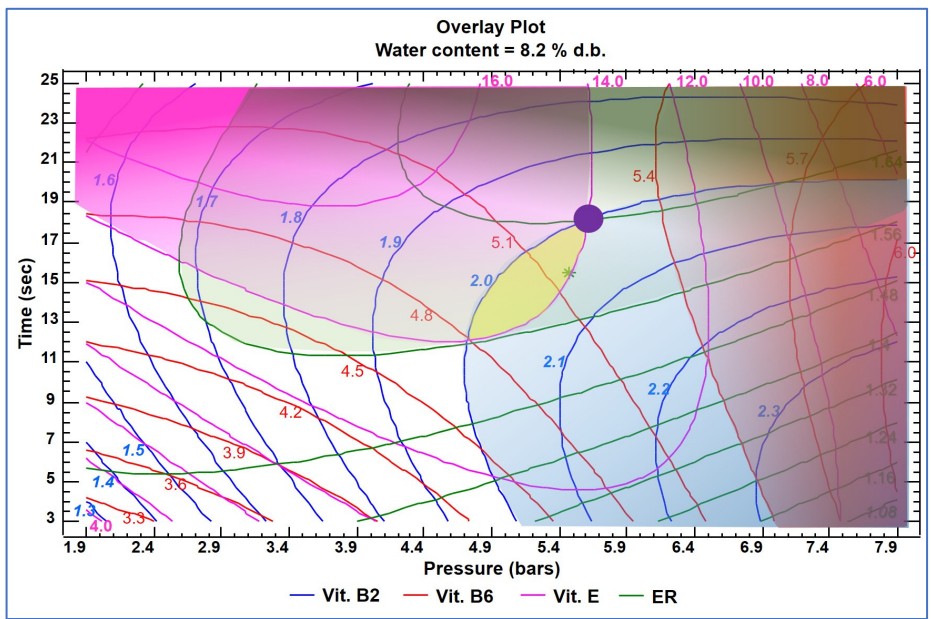

**Figure 7.** Contour plots and optimum areas generated from single and multiple response analysis for ER, Vit.B2, Vit.B6, and Vit.E.

- **Model validation:** To validate the model defined by the software, three repetitions of the IVDV treatment were conducted under conditions of total multiple optimization (Table 4). The values of various responses were then compared to those predicted by the model. Table 5 compiles the predicted results, experimental mean values, and the absolute error for each of the responses studied. Given that the predicted and observed values varied within the margin of the average absolute error, we can conclude that the defined model is consistent with the experimental results, and the equations effectively model the process.

**Table 5.** Model validation.

| Response Parameter | Predicted Value | Observed Value | Mean Absolute Error |
|---|---|---|---|
| ER (%) | 1.6 | 1.58 | 0.02 |
| Vit.B2 (mg/kg) | 2.05 | 2.03 | 0.03 |
| Vit.B6 (mg/kg) | 5.02 | 5.11 | 0.17 |
| Vit.E (mg/kg) | 14.08 | 14.23 | 0.38 |

*3.4. Drying Kinetics*

Comparing the drying kinetics of conventionally hot-air dried sprouts (HAD) to those of treated dried sprouts (HAD + IVDV) confirmed the above findings.

As shown in the drying kinetics in Figure 8, achieving a water content of 5% through conventional hot-air drying required approximately 20 h, with a second phase of approximatively 15 h. Indeed, the compaction of the matrix during this phase of drying made water extraction more challenging, trapping moisture within the material's structure. Moreover, this prolonged exposure to heat involves significant temperature increases within the product, since the energy absorbed by the product could not be used to evaporate the water trapped in the dense matrix. It will function as a sensible heat and will increase the temperature of the product, leading to nutritional content degradation. Hence, conventional hot-air drying substantially degrades the nutritional content, and particularly the vitamin content, of the sprouts.

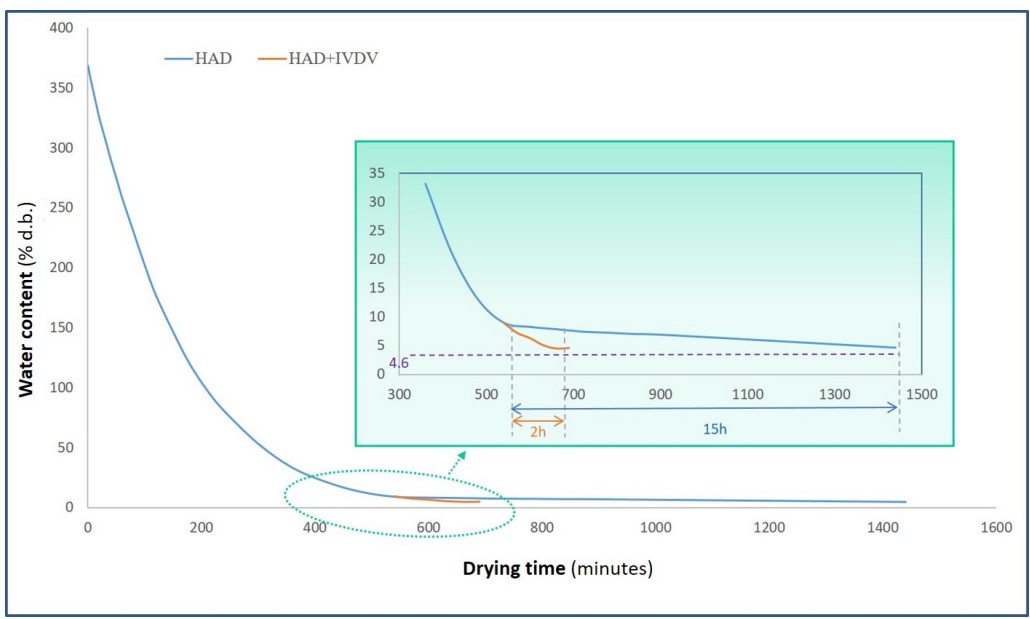

**Figure 8.** Drying kinetics of hot-air dried sprouts (HAD) and IVDV treated sprouts (HAD + IVDV).

Incorporating a texturizing step through IVDV created pores within the product, thereby accelerating material transfer and facilitating the release of the water trapped in the matrix structure. This method remarkably reduced the duration of the final drying phase (from 15 h to 2 h) and lowered the temperature exposure of the product. The energy entering the product is primarily used for the evaporation of the water, which became more available, rather than for heating the product. Consequently, sprouts treated with IVDV and subsequently dried (HAD + IVDV) were exposed to lower temperatures for a shorter duration, compared to conventional drying, leading to better preservation of their nutritional content.

### 3.5. Nutritional and Aspect Comparison of the Treatment Effect on Wheat Sprouts

The evaluation of the different outputs from this work showed that, applied under optimal conditions (pressure of 5.5 bars for a duration of 15.4 s and with a water content of the sprouts before treatment of 8.8%), the IVDV treatment demonstrated improvements in the preservation of vitamins B6 and E, by 412% and 42%, respectively, compared to conventional hot-air drying. Vitamin B2, proteins, and lipids were not significantly impacted by the IVDV treatment. The data in Table 6 below highlight the impact of IVDV on the nutritional content of sprouted seeds by comparing the HAD + IVDV sprouts to ungerminated seeds, fresh sprouts, and HAD sprouts.

**Table 6.** Nutritional content comparison of ungerminated wheat seeds, fresh sprouts, and dried sprouts. Values followed by the same letter within the same column are not significantly different ($p < 5\%$).

|  | Proteins (%) | Lipids (%) | Vit.B2 (mg/kg) | Vit.B6 (mg/kg) | Vit.E (mg/kg) |
|---|---|---|---|---|---|
| Seeds | 9.48 [a] | 2.65 [a] | 0.47 [a] | 2.51 [a] | 6.11 [a] |
| Sprouts | 13.85 [b] | 3.65 [b] | 7.24 [b] | 9.49 [b] | 22.39 [b] |
| HAD + IVDV | 12.33 [c] | 2.97 [a] | 2.03 [c] | 5.11 [c] | 14.23 [c] |
| HAD | 11.90 [c] | 2.77 [a] | 2.19 [c] | 0.98 [d] | 9.91 [d] |

From a visual perspective, it was apparent that sprouts treated with IVDV exhibited a reduction in the green color of their stems, whereas those subjected to conventional drying retained their color more effectively, as depicted in Figure 9. Total hot-air drying of the wheat sprouts resulted in shrinkage and compaction of their grains. In contrast, HAD + IVDV-treated grains retained a shape similar to that of their pre-germination state, along with an increased volume due to expansion. Indeed, expansion is a crucial parameter in the food industry, as it can influence the texture, flavor, and appearance of the final product. The expanded sprouts became crunchy and friable, suggesting new uses for these dried sprouts, such as in breakfast cereal bars, as a crunchy topping that could be sprinkled on salads or soups, or even as an element of animal feed.

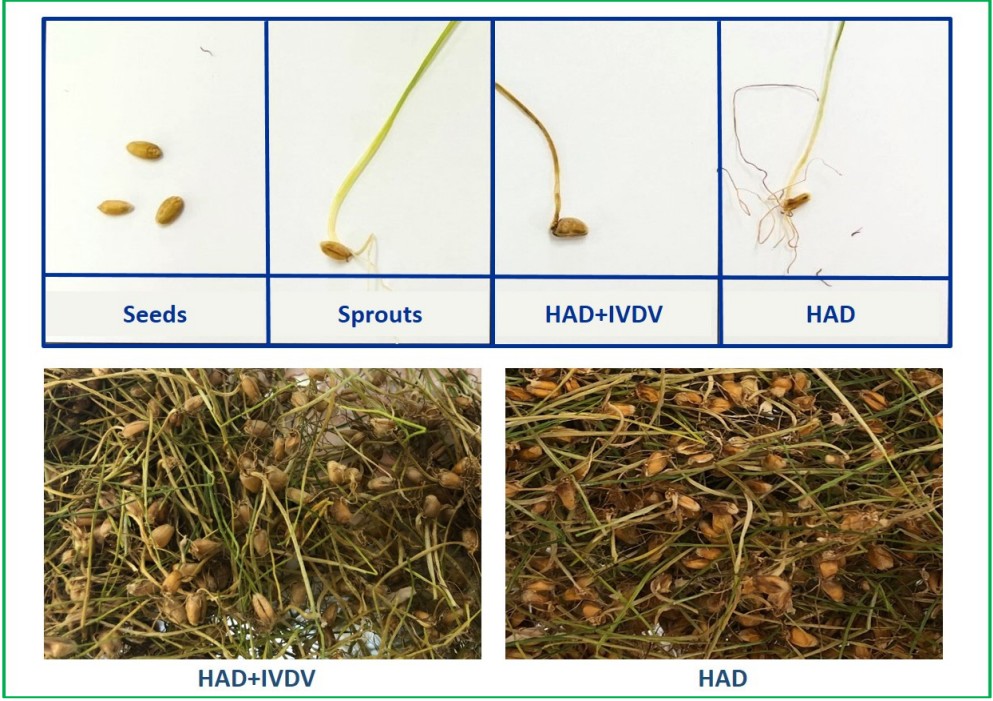

**Figure 9.** Aspect comparison of sprouts.

The amount of research on the drying processes applicable to wheat sprouts is limited within the literature. The influence of air drying on particular attributes of these sprouts has been investigated by a few researchers, such as Hefni and Witthöft [36], who found that the folate levels in wheat sprouts remained stable when subjected to air drying at 50 °C. Similarly, Dziki et al. [37] reported that a mild hot-air drying temperature marginally affected the antioxidant activity of wheat sprouts, while drying at 80 °C decreased it a bit more.

Furthermore, to the best of our knowledge, there have been no comparative analyses on the impacts of various dehydration techniques on the vitamin composition of wheat sprouts. In a related study on broccoli sprouts, Dziki et al. [24] evaluated the outcomes of freeze drying versus air drying, examining variables such as color, phenolic content, and antioxidant capacity. They also assessed the overall drying times, concluding that freeze drying of broccoli sprouts required the longest period. Additional research has explored the effects of grinding wheat sprouts after four days of germination, revealing a reduction in the total drying duration of approximately 50% [23]. However, this method is only feasible when the final product is intended to be flour or powder. Therefore, IVDV is of great interest in terms of drying time.

## 4. Conclusions

The results of this work showed that the optimal IVDV treatment of wheat sprouts is performed at a pressure of 5.5 bars for a duration of 15.4 s, with a water content of 8.8% in the sprouts before the IVDV treatment. Applied under these optimal conditions, the IVDV treatment allowed improvements in the preservation of vitamins B6 and E by 412% and 42%, respectively, compared to conventional hot-air drying, without affecting the contents of vitamin B6, proteins, or lipids. Additionally, achieving a water content of 4.9% through this treatment ensures a longer shelf life for the product and secures its safety for consumption.

The IVDV treatment also led to the sprouts' expansion, resulting in a friable texture and accelerating the drying process. Coupling IVDV with hot-air drying shortened the drying time and, consequently, reduced energy consumption, which was particularly noticeable during the second phase of drying. Therefore, this method presents as a more sustainable and cost-effective alternative to traditional dehydration methods like freeze drying.

Finally, these dried sprouts could find applications in various healthy food products, such as breakfast cereals, cereal bars, and chocolate. The incorporation of sprouted wheat flour into cereal products improves the functional and nutritional quality of these products. Due to their vitamin content (as well as their phenolic content and antioxidant activity, which we have not presented in this study), extracts from these sprouted seeds could be utilized in nutraceutical products as natural antioxidant molecules. Furthermore, the potential benefits of utilizing this process to enhance and preserve the quality of forages warrant attention.

## 5. Patent

Debs, E., Rajha, H. N., Rizk, R., Francis, H., Mrad, R., Maroun, R. G., and Louka, N. (2019). « Dispositif de mise, des installations, sous pression de vapeur élevée avec une Élévation de la Pression Ultra Rapide "ÉPUR". Application à l'Intensification de la Vaporisation par Détente vers le Vide "IVDV" ». Patent No. Ref. n° 2019/05-11666L.

**Author Contributions:** Conceptualization, N.L.; methodology, H.F.; validation, N.L. and E.D.; formal analysis, H.F.; investigation, H.F.; resources, R.G.M. and N.L.; data curation, H.F.; writing—original draft preparation, H.F.; writing—review and editing, E.D., N.L. and R.G.M.; visualization, H.F.; supervision, E.D., R.G.M. and N.L.; project administration, N.L.; funding acquisition, N.L. All authors have read and agreed to the published version of the manuscript.

**Funding:** This research was funded by the Research Council of Saint Joseph University of Beirut (FS68 project) and the Center for Innovation and Technology (CIT).

**Institutional Review Board Statement:** Not applicable.

**Data Availability Statement:** The data supporting the findings of this study can be obtained from the corresponding author upon a reasonable request.

**Conflicts of Interest:** The authors declare no conflicts of interest. The funders had no role in the design of the study; in the collection, analyses, or interpretation of data; in the writing of the manuscript; or in the decision to publish the results.

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
