# Peer review of "Enhancing Wheat Sprout Attributes Using “Intensification of Vaporization by Decompression to the Vacuum”, an Innovative Drying–Texturizing Technology"

_agriculture, doi:10.3390/agriculture14040515_

Round 1
Reviewer 1 Report
Comments and Suggestions for Authors
Comments to authors
The article “Enhancing wheat sprouts attributes and safety using “Intensification of Vaporization by Decompression to the Vacuum”, an innovative drying‐texturizing technology” presents a study on the use of a drying‐texturing system called by the authors Intensification of Vaporization by Decompression to the Vacuum (IVDV) in drying wheat sprouts. The authors studied variables of the IVDV process (saturated vapor pressure, processing time and moisture content) and the effect of the IVDV process + convective drying on vitamins, proteins, lipids and optimization of the expansion rate using response surface methodology.
Dear authors,
I had the opportunity to review your article and would like to draw your attention to some areas where some corrections might be beneficial. Please understand that this feedback is intended to help improve the overall quality of your work. I believe that addressing these points will contribute to the strength of your article.
Below are the observations and improvements needed:
Line 16 – [“Developed in our laboratories…”] this information is not necessary.
Lines 33-34 – It would be interesting to include the values (or range of values) of the proximate composition.
Line 54 – Regarding [“…eading to a reduced average shelf life of approximately 3 to 5 days”] I wonder under what storage conditions.
Lines 69-70 – Regarding [“Additional studies have explored various temperatures in hot air drying processes…”] I wonder which studies? Regarding [“...and only one study has undertaken a comparison between hot air drying and freeze‐drying”] this is not completely true, see for example the following references: https://doi.org/10.1002/jsfa.9630 and https://doi.org/10.35219/foodtechnology.2022.1.07.
Lines 112-116 – I wonder if no sanitary measures were taken to inhibit the development of microorganisms during germination for 7 days.
Line 117 – The subsection “2.3 Hot air drying and IVDV treatment” would benefit from more detailed information on the initial conditions of the sprouts, such as their moisture content immediately after the germination process, the sprout mass used in the partial drying process and in the treatment by IVDV. The authors need to make it clear in the text that the sprout is the product formed by the germinated seed and the radicle that develops during the germination process. This information is essential if another research group is interested in reproducing these results.
Line 143- What is the temperature of water vapor?
Lines 184-186 - In the convection drying process, several parameters are important to ensure drying efficiency: metal mesh dimensions, hole diameter, metal mesh hole area, number of metal mesh holes. Provide information about these parameters, as they may influence the convective mass transfer coefficient to a lesser or greater extent. Also, what format did you use metal mesh in? I imagine it was rectangular. What was the initial mass of the wheat sprouts?
Lines 194-195 – What criteria were used to select the levels of independent variables?
Line 196 – Regarding [“... Every assay is performed in triplicate…”], as the analyzes were performed in three replications, I do not see why the results in Table 1 cannot be presented in the mean ± standard deviation format.
Lines 232-233 – Note that as the interaction between time and moisture content (t×w) did not have a significant effect on the expansion rate (Figure 4a), the term that represents this interaction could have been eliminated from the regression model equation ( Figure 4d) without any prejudice to the significance of the model equation. This makes it more practical to work with these models, especially in the cases of the regression model equations for vitamins B2, B6 and E (Figure 5), which had several terms that did not show significant effects.
Line 256 – The protein content in treated wheat sprouts varied between 10% and 13% ? Não seria entre 10.09% (trial 15) a 12.69% (trial 1)? Furthermore, throughout the text, use only one of the terms, either run or trial.
Lines 443-445 – In Table 1, the results must be presented in the format of mean ± standard deviation. Furthermore, statistical tests must be carried out to verify whether the differences between the different treatments are significant or not.
Line 486 – Regarding [“...but also cut energy use, especially in the second phase of drying”], the authors do not mention in the text the performance of an energy analysis of the proposed drying system (partial drying + IVDV + complementary drying). This statement is indirectly based on the analysis of drying time. Naturally, if the dryer works with the same power, a reduction in drying time results in a reduction in total energy consumption, however, this needs to be clear in the text.
Author Response
The authors appreciate the thoughtful suggestions of the reviewers, which have significantly improved the quality of the manuscript. All the recommendations were carefully considered. Please find below a detailed list of point-by-point responses.

Reviewer 2 Report
Comments and Suggestions for Authors
The presented manuscript, "Enhancing Wheat Sprouts Attributes and Safety using Intensification of Vaporization by Decompression to Vacuum," describes an interesting process for wheat sprout preservation. The topic is relevant and offers valuable insights for readers. I recommend it for publication after minor revisions. Below are some comments to improve the manuscript:
Title: The title paper explains the manuscript's topic, however the inclusion of the word "safety" may imply a focus on food safety. I suggest to remove the word "safety" from the title. On the other hand, the authors could include information about the reduction of microbial patogens using the IVDV process.
Abstract: The abstract provides a clear overview of the study on the Intensification of Vaporization by Decompression to Vacuum and the importance of wheat sprouts. The methodology and results are correctly presented.
Introduction: The introduction offers a comprehensive overview of the study's background, context, and objectives. It is well-structured, discussing the importance of wheat sprouts, their nutritional content, and traditional processes. The introduction appropriately references several studies to support statements.
While reviewing a similar process, I found the Flash détente process.
Food Reviews International, 40:3, 833-866, DOI: 10.1080/87559129.2023.2197997
https://agris.fao.org/search/en/providers/122439/records/6472332e53aa8c89630206c7
Please, could the authors mention if the process is similar or what is the difference?
Material and Method section: This section offers a detailed description of the experimental setup. It is suggested to specify the units used for reporting water content, whether it is on a dry basis or wet basis.
Results and Discussion section: This section provides a comprehensive analysis of wheat sprout attributes and optimization of process parameters. It is well-organized and offers a clear overview of the study's findings.
Section 3.1.2 and 3.2.2: Please include a discussion on the differing behaviors of vitamin B2 compared to Vitamins B6 and E, potentially considering the chemical composition.
In Figure 6, please add units to the axes.
In Figure 8, the y-axis unit is water content %, whereas in Section 2.5, the equation is presented as a ratio between mass water sample at time "t" and mass of dry matter. Please specify the correct units in the equation or results.
Conclusion section: The conclusion is well-written and summarizes the main points of the manuscript effectively.
References: It is recommended for the authors to reduce self-citation in the references.
Author Response

(The authors gave the same response as above.)

Reviewer 3 Report
Comments and Suggestions for Authors
General Comments
Title and in the entire document; The name of the new process doesn't seem to be well-refined and needs modification. Suggestions: "Intensified Vaporization by Decompressed Vacuum (IVDV)" or "Decompressed Vacuum-Intensified Vaporization (DVIV)", ... or whichever way it makes clear sense. The name should give a good and clear clue to the reader.
Conceptual: The vacuum drying might still create a conducive environment for anaerobic psychrotrophic pathogens that might survive the freezing phase and become active later. Is this not an issue of concern? The authors might need to give some insights here.
Materials and Methods - Design information; The experimental design is perfect and clear. But, the authors are advised to present the coded (-1, 0, & +1) and actual (low, middle, high) levels of the independent variables in a table form. Also mentioning the basis for the selected levels of each variable is cruicial. Suggestion made as follows:
|
Variables |
Low |
Medium |
High |
|||
|
Coded |
Actual |
Coded |
Actual |
Coded |
Actual |
|
|
Pressure (bars) |
-1 |
3 |
0 |
4.5 |
+1 |
6 |
|
Time (sec) |
-1 |
7 |
0 |
12 |
+1 |
17 |
|
Moisture (%) |
-1 |
15 |
0 |
25 |
+1 |
35 |
Also, the experimental design and data analysis approach for the comparison between HAD+IVDV versus HAD alone (Table 4 results - Categorical variables - needing ANOVA) is missing from the section and should be included.
Results and Discussions, an important summary table is needed on the is missing on the significance (p-value) of the Lack of fit information and the goodness of fit (Coefficient of Determination, R2 & R2adj. values or any other indicatgor of the goodness of fit). With out these information, it is totally incomplete to talk about the Tables 1 data and the bar and surface+contour graphs presented.
The bar graphs presented along side the surface graphs are not clear. Q0. What are the legends indicating with the "+"/"-" signs?
Specific Comments
Line 27: remove the word "interesting"
Line 32: language, change "one" to "among" for subject-verb agreement (are vs one); and also the entire statement "... food trends expected to continue growing in the coming years ..." growing in terms of what? use demand for instance.
Line 33: Change "They" (vague) to a specific noun (Sprouts),
Line 34: subject-verb incompatibility, "... sprouts are a good source..." are (plural) against "a good" (singular), a lot of language issues throughout the document.
Lines 40-41: A stronger justification is needed as to why wheat. Wheat is a major staple crop in the grain form and sprout production should not affect the global food security.
Line 50: change the last part of the statement to "in their fresh form"
Lines 60-61: the term "drying" is repeated in the same statement and either the statement needs to be rephrased or the second "drying" is cut out. The same issue is under line 71, where the first appearance is not needed.
Line 76, "stage" is repeated in the same statement.
Line 82: change "flour" to "powder", flour is typically for grains
Lines 83-84: the patent should be cited.
Lines 86-89: repetition of words in the same statement; "pressurization", "high pressure" (2x), "second(s)"
Lines 95-96: more repetitions, "heat sensitive products"
Lines 109-116: It is not clear how the end point of the sprouting was determined. Q1. How tall was/were the root/shoot? It should be clear.
Line 120: The Pre-IVDV moisture level should be mentioned in parentheses.
Line 142: the term "intimate" is confusing and should either be changed or removed.
Lines 152-153: the part that reads "... simultaneous injection of atmospheric air while maintaining the vacuum." doesn't make any practical sense. Ther simplest concept is that, if there is vacuum, there is no atmospheric air or vice versa. Q2. What exactly is happening at this stage? Is is a lo pressure/partial vacuum situation? if so it must be stated as such, NOT vauum plus atmospheric air coexisting at all.
Line 172: the term "assay" is confusing and should be removed.
Line 185: the specific interval should be clearly mentioned.
Lines 196-197: repeatitions/replications are not needed except for the center points in RSM designs. Q3. Why did the authors want to do the replications? How was the data handled in the modelling analysis?
Line 270: the R2 values and the lack of fit information should be summarized in a separate Table.
Line 285: spacing needed between the citation and the nbext word "with"
Line 329: citation of figure and number is neeeded in parenthesis next to the Pareto Diagrams.
Line 351: Synergy is when the effect is multiple folds greater than the individual variables and their summations and, when it is negative, (like on Vit. B2 and E) mentioned in the same paragraph, it is called "antagonasitic". Authors need to check and correct the discussions here and use the correct terminologies, likely "interactions" as mentioned in the topic title.
Line 437: insert "... data in ..." betweeb "The" and "Table 4" at the beginning of the new statement, begin Table with upper letter "T".
Lines 489-496: the last paragraph of the Conclusion section is exotic to the work and is not based on the research data. It would be great if that part is removed.
Comments on the Quality of English Language
English can be improved. Some terms/words are exotic to contexts and some are repeated in the same statements.
Author Response

(The authors gave the same response as above.)

Round 2
Reviewer 1 Report
Comments and Suggestions for Authors
Although they present an improved version of the article, the authors clearly avoided more laborious revisions. Additionally, simple things requested such as including the accuracy of your measurements were not included in the revised version of the article. Furthermore, I have no further comments.
Reviewer 3 Report
Comments and Suggestions for Authors
The authors have improved the write-up and provided explanations for those comments they didn't consider. The article is good for publication.